# Fluorescent Aromatic Polyether Sulfones: Processable, Scalable, Efficient, and Stable Polymer Emitters and Their Single-Layer Polymer Light-Emitting Diodes

**DOI:** 10.3390/nano14151246

**Published:** 2024-07-25

**Authors:** Konstantinos C. Andrikopoulos, Despoina Tselekidou, Charalampos Anastasopoulos, Kyparisis Papadopoulos, Vasileios Kyriazopoulos, Stergios Logothetidis, Joannis K. Kallitsis, Maria Gioti, Aikaterini K. Andreopoulou

**Affiliations:** 1Department of Chemistry, University of Patras, University Campus, 26504 Rio-Patras, Greecexanastasop@upatras.gr (C.A.);; 2Nanotechnology Laboratory LTFN, Department of Physics, Aristotle University of Thessaloniki, 54124 Thessaloniki, Greece; detselek@physics.auth.gr (D.T.); logot@auth.gr (S.L.); 3Organic Electronic Technologies P.C. (OET), 20th KM Thessaloniki—Tagarades, 57001 Thermi, Greece; 4Foundation for Research and Technology Hellas/Institute of Chemical Engineering Sciences (FORTH/ICE-HT), Platani Str., 26504 Patras, Greece

**Keywords:** light-emitting polymers, photoluminescence, polyethersulfones, electroluminescence, PLEDs, single-layer light-emitting devices, white light emission

## Abstract

In this study, fully aromatic polyether sulfones were developed, bearing blue, yellow, and orange–red π-conjugated semiconducting units. Carbazole-, anthracene-, and benzothiadiazole-based fluorophores are copolymerized with a diphenylsulfone moiety. A diphenylpyridine comonomer was additionally utilized, acting as both a solubilizing unit and a weak blue fluorescent group. Using this rationale, fluorescent polyarylethers with high molecular weights, up to 70 kDa, were developed, showing film formation ability and high thermal stability, while preserving excellent solubility in common organic, nonvolatile, and nonchlorinated solvents. Fine-tuning of the emission color was achieved through subtle changes of the comonomers’ type and ratio. Single-chromophore-bearing copolymers emitted in the blue or the yellow region of the visible spectrum, while the dual-chromophore-bearing terpolymers emitted throughout the visible spectrum, resulting in white light emission. Solutions of 20 wt% in polar aprotic solvents at ambient conditions allowed the deposition of fluorescent copolyethers and printing from non-chlorinated solvents. All polyethers were evaluated for their structural and optoelectronic properties, and selected copolymers were successfully used in the emitting layer (EML) of organic light-emitting diode (OLED) devices, using either rigid or flexible substrates. Remarkable color stability was displayed in all cases for up to 15 V of bias voltage. The Commission Internationale de L’Eclairage (CIE) of the fabricated devices is located in the blue (0.16, 0.16), yellow (0.44, 0.50), or white region of the visible spectrum (0.33, 0.38) with minimal changes according to the ratio of the comonomers. The versatile methodology toward semiconducting polyethersulfones for polymer light-emitting diodes (PLEDs) developed herein led to the scaled-up production of luminescent polymers of up to 25 g of high-molecular-weight single batches, demonstrating the effectiveness of this approach as a straightforward tool to facilitate the synthesis of flexible and printable EMLs for large-area PLED coverage.

## 1. Introduction

Polymer light-emitting diodes (PLEDs) are gaining growing interest due to their inherent advantages compared to their inorganic, small, organic-based LED counterparts. Among these advantages are being lightweight, having a comparatively simple device structure; being cost-effective, easy to fabricate, and printable; and requiring only a single emitting layer [1,2,3]. In particular, solution processing is an essential merit of PLEDs, both in the research and industry sectors, for large-area, flexible, foldable, and low-cost displays [4,5,6], allowing the deposition of the EML via thin-film forming techniques such as spin coating [7], blade coating [8], spray coating [9], roll-to-roll processing [10], and inkjet printing [11,12].

Semiconducting polymers are attractive candidates for the emitting layer (EML) of LEDs, as they generally have simple synthetic procedures, hassle-free purification procedures, structural diversity, and integrated functionalities [13,14,15,16]. Depending on their molecular weights, they offer high thermal and chemical stability and excellent film-forming properties, making them ideal for solution processing [17]. The most prominent examples of the polymers used in the EMLs include fully aromatic conjugated polymer structures prepared typically via catalyst-mediated coupling polymerizations [18], such as Suzuki, Heck, or Stille coupling polycondensations. As such, benchmark luminescent polymers are the blue-emitting polyfluorenes [19,20] (PFN, PFO) and the yellow polyphenylene-vinylenes (“Super-Yellow”) [21,22,23]. Such fully conjugated polymers, however, present difficulties in their purification from catalyst traces and often exhibit differences in their structural and optoelectronic properties due to the differences in the molecular weights of different batches [24]. In addition, even though red- and yellow-green-emitting polymers exhibit satisfactory performance, the acquisition of high-performance blue-emitting polymers remains a challenge due to the high charge injection barrier and unbalanced charge injection and transportation, owing to their intrinsic wide band gap [25,26]. 

As an alternative approach, poly (arylene ether)s [27,28,29] and rigid–flexible aromatic–aliphatic polyethers have been proposed as EML candidates [30]. In these cases, the rigid conjugated moieties impart light-emitting properties to the polymers while the ether linkages and the flexible aliphatic spacers help to improve the solubility and processability of the polymers. The adjacent fluorophores, when separated, are dispersed along the polymeric backbone, leading to enhanced light emission of the polymers in the solid state. Furthermore, their fixed conjugation length results in optoelectronic properties that remain constant for the different synthesized batches, and their catalyst-free polymerization approaches lead to simpler purification procedures [31,32]. However, solubility issues may arise when higher molecular weights are achieved in several cases of aromatic–aliphatic, rigid–flexible luminescent polyethers. As a result, it may be necessary to use strong, high-boiling-point, chlorinated solvents like chlorobenzene and o-dichlorobenzene to deposit them as EMLs in PLEDs [30,32].

To address the solubility issues mentioned above, using aromatic fluorescent polyethersulfones shows promise. Here, the effective conjugation length of organic chromophores can be combined with the electron-accepting nature of diphenylsulfone [33,34]. This has been recently shown as a proof of concept with fluorescent polyethersulfones showing adequate molecular weights and solubility compared to the previously studied aromatic–aliphatic polyethers, allowing the preparation of stable inks used for the fabrication of solution-processed white OLEDs (WOLEDs) with a single EML structure [31]. White light emission from single EMLs via solution printing techniques offers great advantages over multicomponent EMLs, in which the complicated device structure leads to unstable emitting color. White light single EML PLEDs are of utmost significance for future artificial lighting applications to reduce the energy consumption and carbon emissions of incandescent bulbs and fluorescent tubes [35,36,37].

The study reported herein focused on the extensive and in-depth investigation of fluorescent aromatic polyether sulfones and their device characteristics for single-color PLEDs and single EML WPLEDs with high efficiency, stability, and printability. Distyrylanthracene, distyrylcarbazole, and distyrylbenzothiadiazole chromophores in the form of yellow, blue, and orange–red emitters, together with the electron-accepting nature of diphenylsulfone, were utilized for the development of fully aromatic fluorescent polyethersulfones. The fluorescent polyethersulfones reported here emit light at specific wavelengths in the visible spectrum, based on the type of chromophore, or they can emit white light when different chromophores are combined along single polymeric backbones. A diphenylpyridine comonomer was also incorporated alongside the fluorescent monomers in copolymeric or terpolymeric structures, which further improved the solubility of the materials, their fluorescence properties, and color purity by increasing the distance between adjacent chromophores. Overall, these fluorescent aromatic polyether sulfones possess all the necessary characteristics as light-emitting polymers for large-area, printed, and flexible PLEDs. 

## 2. Materials and Methods

### 2.1. Materials

Chemicals and reagents were purchased from commercial providers and used without further purification. Triethylamine was distilled over CaH_2_, and N,N-Dimethyl formamide (DMF) was dried over molecular sieves (4 Å). 

The homopolymer (pySO_2_) [38] and 9,10-dibromo-anthracene [39] were prepared according to literature procedures. Monomers 9,10-bis (p-acetoxystyryl)anthracene (Anthr) [32], 2,7-bis (p-acetoxystyryl)-9-(2-ethylhexyl)-9H-carbazole (Cz) [30,32], 4,7-bis (p-acetoxystyryl)-2,1,3-benzothiadiazole (Bz) [30] and 2,6-bis(4-hydroxyphenyl)pyridine (HOpy) [38] have appeared previously in the literature. In this work, rigorous optimizations took place for the yields and quantities of the monomers. Detailed experimental procedures and characterization data are given in the Appendix A.

### 2.2. Instrumentation

For most monomer synthesis, a 10 L single jacked borosilicate Radleys Reactor-Ready^TM^ Pilot (Radleys, Shire Hill, Saffron Walden, Essex, UK) was used equipped with a PTFE Anchor Stirrer Shaft attached onto a Hei-TORQUE 200 Precision overhead stirrer (Heidolph Instruments GmbH & Co. KG, Schwabach, Germany) and a Pt100 PTFE temperature probe (Bohlender GmbH, Grünsfeld, Germany). A Huber Unistat 705 thermoregulator (Peter Huber Kältemaschinenbau SE, Offenburg, Germany) with an operating temperature range from −75 °C to 250 °C was connected to the pilot reactor. 

^1^H, ^13^C Nuclear Magnetic Resonance (NMR) spectra were recorded on a Bruker Advance (Bruker BioSpin GmbH, Magnet Division, Karlsruhe, Germany) DPX 600.13 and 150.90 MHz spectrometer, respectively, with CDCl_3_ as solvent containing Tetramethyl silane (TMS) as internal standard.

Gel permeation chromatography (GPC) measurements were carried out using a Polymer Lab chromatographer (Agilent Technologies, Santa Clara, CA, USA) equipped with two PLgel 5 μm mixed columns and a UV detector, using CHCl_3_ as eluent with a flow rate of 1 mL/min at 25 °C calibrated versus polystyrene standards. All samples were filtered via PTFE membrane filters of 0.45 μm and 0.2 μm pores before injection to the GPC columns. 

Thermogravimetric analysis (TGA) was carried out on ~8 mg samples contained in alumina crucibles in a Labsys TM TG apparatus of Setaram (Caluire, France) under nitrogen and at a heating rate of 10 °C/min. 

UV-Vis spectra were recorded using a Hitachi U-1800 spectrophotometer (Hitachi High-Technologies Europe GmbH, Mannheim, Germany). Continuous wave photoluminescence was measured on a Perkin Elmer LS50B spectrofluorometer (Waltham, MA, USA). All UV-Vis and PL measurements were performed in air using quartz cuvettes and flat quartz substrates for the examination of solutions and films, respectively. 

The electrochemical behavior of the fabricated materials was investigated using cyclic voltammetry (CV). CV experiments were carried out in a three-electrode cell. A fluorine-doped tin oxide (FTO) substrate was used as the working electrode, a Pt wire was used as the secondary electrode, and a saturated Ag/AgCl reference electrode was used in the cell. Thin films of the fabricated materials were drop-cast on FTO-coated glass slides, (R_sheet_~8 Ω/square) and preheated at 80 °C for 20 min, from precursor chloroform solutions. The resulting films were further annealed at 80 °C for 15 min. An Autolab PGSTAT 302 N electrochemical analyzer (Metrohm AG, Herisau, Switzerland) connected to a personal computer running the NOVA 1.8 (Metrohm AG, Herisau, Switzerland) software was used for data collection and analysis. All experiments were carried out at a scan rate of 0.1 V/s. Tetrabutylammonium hexafluorophoshate (TBAPF_6_) 0.1 M of anhydrous acetonitrile (CH_3_CN) was used as supporting electrolyte. Before carrying out the measurements, the cell was purged with pure argon for 20 min to remove diluted gases. The reference electrode potential was calibrated against Ferrocene/Ferrocenium (Fc/Fc^+^) after each voltammetry run. 

The Highest Occupied Molecular Orbital (HOMO) energy levels were calculated from the first reduction onset potential using the equation [40,41]:(1)$$E_{\mathrm{HOMO}}=e\;(E_{on}^{OXID}-E_{1/2}^{Ferrocene})-5.2[\mathrm{eV}]$$

$E_{on}^{OXID}$ = the onset determined for the oxidation peak of each molecule in cyclic voltammetry (V) versus Ag/AgCl.
$${E\;(}_{1/2}^{Ferrocene})=(E_{oxid}+E_{OX})/2\;vs.\;Ag/AgCl$$

The Lowest Unoccupied Molecular Orbital (LUMO) energy levels were calculated from the equation:(2)$$E_{g}^{opt}\;=\;\mathrm{HOMO}-\mathrm{LUMO}$$

The $E_{g}^{opt}$ (optical band gap) was determined from the equation:(3)$$E_{g}^{opt}=\frac{1240[eV\;nm]}{\lambda onset[nm]}\;[eV]$$

The value of −5.2 eV in Equation (1) emerged from the calibration of the Ag/AgCl electrode versus a normal hydrogen electrode (NHE) and considering that the NHE redox potential corresponds to −4.6 eV on the zero vacuum level scale [42,43].

The spin coating method was implemented for the development of the poly-3,4-ethylene dioxythiophene: polystyrene sulfonate PEDOT:PSS layer, acting as the Hole Transport Layer (HTL) as well as the emitting layer, to produce OLED devices on pre-patterned indium-tin oxide (ITO)-coated glass substrates, in the glove box. A mini roller coater (FOM Technologies, Kastrup, Denmark) was used to develop, by the slot die process, the PEDOT:PSS and emitting layers onto PET/ITO flexible substrates. In both cases, Ca and Ag layers were used as a cathode electrode bilayer and deposited using appropriate shadow masks through Vacuum Thermal Evaporation (VTE).

For the HTL, a solution of PEDOT: PSS (purchased from Clevios Heraus, Hanau, Germany) Clevios P VP AI 4083 mixed with ethanol in the ratio of 2:1 was prepared either for spin coating or the slot die process. In the case of emitting films, the concentration of blue-emitting polymers was equal to 1% (*w*/*v*) for the spin coating deposition, whereas for the slot die coating, it was 4% (*w*/*v*). For the yellow-emitting polymers, the solution concentration was 4% (*w*/*v*) for both spin coating and slot die coating. Finally, the resulting solution concentration of all studied white-emitting polymers was 1% (*w*/*v*). For the case of spin coating, the HTL films were formed using 4500 rpm, whereas all the emitting films were formed using 2000 rpm. Finally, the PEDOT:PSS and the emitting films were formed using the slot die process with the optimized deposition parameters in each case. The aim was to achieve thicknesses of 45–55 nm and 50–80 nm, respectively.

The electroluminescence measurements (EL) of the PLED devices were performed using the External Quantum Efficiency (EQE) system (C9920-12) (Joko-cho, Higashi-ku, Hamamatsu City, Japan).

## 3. Results

### 3.1. Synthesis and Characterization of Monomers

In our earlier reports on the development of light-emitting polymers [30], we utilized flexible aliphatic spacers as conjugation break units between the π-conjugated segments. This approach was effective and adaptable for controlling the emission color of the resulting light-emitting polymers. It also facilitated the production of high-molecular-weight polymers with excellent film-forming properties and mechanical stability. Additionally, it ensured the solubility and processability of the polymers in most instances. However, scalability and improved processability at ambient environmental conditions remained open issues. Especially when large-area flexible OLEDs are the targeted application, several distinct prerequisites must be simultaneously fulfilled. These are high molecular weights that assure active layers’ integrity and stability over time and temperature deviations, enhanced solubilities [30,32] in non-volatile solvents so that polymer inks remain stable regardless of time and storage conditions, and optimized synthetic protocols offering scalability in single-batch multigram quantities.

Therefore, an alternative pathway is herein adopted for developing versatile, processable, and scalable light-emitting fluorescent polymers, ultimately targeting printable large-area OLEDs. Via this route, difunctional fluorescent monomers bearing the two acetoxystyryl groups are scaled up in multigram amounts and are readily polymerized under high-temperature polyetherification condensation conditions, affording stable and scalable polyethersulfones. These chromophore monomers ensure multigram single-batch polymerization reactions, creating consistent light-emitting polymers for the active layer of OLEDs of repeatable and constant properties and efficiencies.

The general synthetic pathway for the fluorescent chromophore monomers prepared in this study is presented in Appendix A. This methodology takes advantage of the minimum effective conjugation length required to create chromophores emitting at specific wavelengths of the visible spectrum. As such, monomers bearing single anthracene (Anthr), carbazole (Cz), or benzothiadiazole (Bz) cores were selected to provide yellow, blue, or orange–reddish emissions, respectively. Based on our previous findings, 9,10-dibromo-anthracene was used as the yellow core due to its excellent optoelectronic properties accompanied by high thermal and fluorescence stability [44]. As the blue fluorophore, a carbazole core, due to its strong electron-donating ability and high-energy triplet states, bearing solubilizing N-ethylhexyl side chains, was prepared, starting from the N-alkylation at the 9H position of 2,7 dibromo 9H carbazole. As an orange–red emitter, a dibromo-benzothiadiazole core was employed due to its good semiconducting properties as well as high thermal stability and chemical tolerance [44]. The three chromophores reacted with 4-acetoxy styrene under Heck coupling conditions, leading to the final polymerizable 9,10-bis (p-acetoxystyryl)anthracene (Anthr), 2,7-bis (p-acetoxystyryl)-9-(2-ethylhexyl)-9H-carbazole (Cz) and 4,7-bis (p-acetoxystyryl)-2,1,3-benzothiadiazole (Bz), as shown in Appendix A. Optimization of the Heck coupling reaction conditions was performed in terms of catalyst loading, temperature, and duration, affording the Anthr, Cz, and Bz final diacetoxy-monomers in 48 g, 32 g, and 5 g single-batch quantities, respectively, as can be seen in the Experimental Section of the Appendix A. The ^1^H NMR spectra of the synthesized monomers are provided in Appendix A. 

### 3.2. Synthesis and Characterization of Polymers

Moving on to the synthesis of the fully aromatic luminescent polymers, the chromophore monomers of Appendix A, together with a pyridine-bearing diphenyl diol, namely 2,6-bis (4-hydroxyphenyl)pyridine (HOpy), and with bis (4-fluorophenyl)sulfone (diFSO_2_) were combined under different modes and ratios, as depicted in Figure 1. High-temperature polyetherification condensation conditions were employed in a high-boiling-point non-protic solvent (DMAc) using potassium carbonate as the base. Toluene was added in the copolymer and terpolymer cases to create azeotropic mixture with the water formed during the polymerization of the HOpy carrying the two hydroxyl groups, and this further aids the kinetics of the reaction. The difluoro-phenyl-sulfone (diFSO_2_) monomer was selected among all other aromatic difluorides, since it is known to afford high-molecular-weight aromatic polyethers, imposes thermal, chemical, and oxidative stability [34,45], and improves the solubility of the final materials, allowing processability and film formation via solution casting. Additionally, the electron-accepting sulfone moieties (HOMO = −7.11 eV, LUMO = −1.37 eV) can enhance the charge-transporting properties of the corresponding semiconducting aromatic polyethers [33].

Starting from the semiconducting homopolymers bearing carbazole (CzHom) (Figure 1a) or anthracene (AnthrHom) (Figure 1b) chromophore cores, these were prepared using equimolar amounts of the luminescent distyryl monomers Cz or Anthr, respectively, and difluorophenyl sulfone (diFSO_2_). However, the CzHom homopolymer exhibited moderate solubilities in chlorinated solvents but superior solubilities in non-chlorinated, non-protic solvents like N-methylpyrrolidone (NMP), N,N-dimethylacetamide (DMAc), or DMF, even in ambient conditions. Therefore, the molecular weight of CzHom was severely underestimated via GPC using CHCl_3_ as the mobile phase, since only the lower-molecular-weight fractions could be solubilized in CHCl_3_, whereas all high-molecular-weight fractions remained insoluble. To surpass the low solubility of this homopolymer, the insertion of the HOpy moiety along the semiconducting backbones was chosen to improve the processability of the luminescent polymers from solution. In addition, due to its heteroatom nitrogen-bearing phenyl ring, the HOpy diol was expected to enhance the semiconducting character of the polymeric backbones. Also, because of the nitrogen-containing phenyl ring in HOpy diol, it was expected to enhance the semiconducting properties of the polymeric backbones. Furthermore, as it only absorbs and emits within the UV region of the light spectrum, it could serve as a host material, diluting and separating nearby chromophores, as will be discussed below in more detail. 

All copolymers containing the HOpy comonomer exhibited enhanced solubilities in most common organic solvents, chlorinated and non-chlorinated ones like CHCl_3_, DMAc, and NMP. CzCop and AnthrCop luminescent copolymers were prepared under different HOpy-to-chromophore ratios, as depicted in Figure 1a,b and Table 1, to evaluate the effect of the chromophore’s content on the polymers’ optoelectronic properties. Table 1 provides the “best-off” luminescent homo- and co-polymers that showed the highest molecular weights, while the respective GPC curves are shown in Appendix A. The CzCop reached Mn values up to ~68 kDa, while the AnthrCop was up to 70 kDa. These were accomplished after systematic optimizations in terms of polymerization conditions, such as temperature, duration, concentration, and work-up procedure. Tables of all the Cz- and Anthr-based polyether sulfones synthesized in this work are provided in the Supporting Information section Appendix A, revealing in detail the efforts and progress toward optimization and scaling up of the luminescent polymers developed herein. The CzCop and AnthrCop were prepared in single batches and the highest amounts obtained are also shown in Table 1. 

In the quest for white light-emitting polymers that can be readily applied in large flat or curved areas of various shapes, luminescent aromatic polyether sulfones comprising two fluorophores were reported by our group in the first preliminary study of this type of semiconducting polysulfones. In this study, we used the same polymerization method to copolymerize the fluorescent monomers bis (p-acetoxy styryl) carbazole (Cz) and bis (p-acetoxy styryl)benzothiadiazole (Bz) fluorescent monomers together with the HOpy and the diFSO_2_ co-monomers, as illustrated in Figure 1c, aiming to achieve even higher molecular weights and polymerization yields. As has been already shown in our initial report, the chosen Bz and Cz fluorescent monomers, serving as orange–reddish and blue emitters, respectively, provide a wider emission profile across the visible region, resulting in white light emission. Different Cz-to-Bz ratios were employed to fine-tune the optical properties of the final WTP polymers. In all cases, a 50:50 ratio of the luminescent chromophores to the HOpy monomer was maintained (see Table 2, Appendix A). As in the copolymers’ case, high to very high molecular weights were accomplished for the WTPs. Notably, the terpolymers’ solubility in common organic solvents and processability from both lower-boiling-point solvents like CHCl_3_ and higher-boiling-point, non-volatile, non-chlorinated solvents like DMAc, NMP, etc., are preserved. 

The structural characterization of all copolymers and terpolymers was performed via ^1^H NMR spectroscopy (Appendix A).

The thermal stability of the copolymers was monitored using thermogravimetric analysis (TGA) both under air and under nitrogen flows (indicative examples are shown in Appendix A). Under nitrogen, the copolymers are thermally stable up to 400 °C with only a 3%wt weight loss at 160–180 °C due to residual solvent traces and a 62–56%wt residue at 800 °C. Under air, the AnthrCop and CzCop copolymers exhibited a single degradation step starting at 350 °C and 280 °C, respectively. The WTPs exhibited a first degradation at about 175 °C up to 280 °C with a weight loss that increased with the Bz comonomer content, from 3%wt for WTP3 with Bz = 0.25% to 5%wt for WTP2 with Bz = 0.5%wt. Then, three subsequent degradation steps occurred for the WTPs at 325 °C, 490 °C, and 570 °C. Overall, the studied copolymers showed excellent thermal stability, proving that these luminescent aromatic polysulfones can withstand elevated temperatures of at least 180 °C in air and 400 °C in inert gases. 

### 3.3. Optical and Electrochemical Characterizations

To elucidate the photophysical properties of the synthesized luminescent copolymers, their absorption (Appendix A) and emission (Figure 1a–c) profiles were first investigated in solution. The absorption of the blue-emitting copolymers reveals a bimodal peak with maxima at λ = 377 nm and λ = 398 nm, attributed to the Cz moiety and a shoulder at λ = 306 nm, attributed to the HOpy moiety (Appendix A). As expected, the ratio of the intensities between the two peaks correlates to the ratio of the two comonomers in the polymeric chain. When the Cz percentage is increased, it results in a more noticeable double absorption peak. In the case of the yellow-emitting copolymers AnthrCop (Appendix A), an absorption peak at λ = 413 nm attributed to the Anthr moiety is observed in all copolymers. The second shoulder at λ = 314 nm is attributed to the HOpy moiety. Finally, the white-emitting terpolymers (WTPs) (Appendix A) have a similar profile to the blue-emitting copolymers due to the high percentage of the Cz moiety in these polymers. Appendix A’s inset shows a zoomed-in range for λ = 420–540 nm, depicting the absorbance related to the Bz moiety. The intensity ratio is directly linked to the percentage of Bz present in the polymer, as expected. 

Moving on, the absorbance of all polymers in thin film form was investigated (Appendix A). In this case, the vibronic peaks are less pronounced compared to the solution absorbances; however, the general maxima are preserved. The copolymers that emit blue light have an absorbance peak at a wavelength of 339–448 nm for the Cz moiety, which is slightly red-shifted compared to the absorbance in their solution. Additionally, there is a less prominent shoulder at around 300 nm due to the HOpy moiety (Appendix A). The Anthr-based yellow light-emitting polymers exhibit an absorbance peak at λ = 420 nm, slightly red-shifted from the solution absorbance (Appendix A). In the case of the white light-emitting terpolymers (WTPs) (Appendix A), a less pronounced peak at λ = 381 nm, owing to the Cz moiety, is again slightly red-shifted compared to the solution absorption spectra. The absorbance of the Bz moiety at λ = 507 nm has a higher intensity compared to the solution absorbance, likely due to a higher degree of charge transfer from the Cz to the Bz in the more closely packed polymeric chains in the solid state.

The emission profiles of the solutions and of the thin film forms were examined for all polymers, and the results are shown in Figure 1. The blue light-emitting Cz-based copolymers in DMF solutions exhibit a bimodal peak at λ = 424 nm and λ = 443 nm (Figure 1a). The relative intensities of these two peaks are directly related to the percentage of carbazole in the polymer chain, which is depicted in the inset of Figure 1a. In the case of CzHom, the peak at λ = 443 nm has a higher intensity, while for decreasing percentages of carbazole, the peak at λ = 424 nm becomes dominant. The emission profiles of Cz blue light-emitting copolymers in film form (Figure 1d) revealed emissions of less defined structures and broader maxima at λ = 461–470 nm. The difference in the relative intensities of the Cz emission peaks in Figure 1a could be attributed to the different interactions of the chromophore moieties in the solution state. In the film state, all copolymers display an amorphous emission profile. This suggests that in the solution state, the components have the freedom to interact, likely due to the dynamic nature of the system, which is not present in the film state.

In the case of the Anthr-based yellow-emitting copolymers (Figure 1b), their emission in DMF solutions is centered in the yellow–green region at about 485–510 nm. For three of the Anthr-based copolymers, namely the AnthrHom, AnthrCop 70/30, and AnthrCop 30/70, a single peak can be seen at λ = 511 nm. For the AnthrCop 50/50 and AnthrCop 10/90 copolymers, a bimodal peak was observed with maxima at λ = 486 and λ = 509 nm, attributed to different interactions of the Anthr chromophores in solution. The Anthr-based copolymers in film form emit within a wider range, compared to their solutions, depending on the percentage of the Anthr chromophore loading of the copolymer (Figure 1e). The emission profiles of all Anthr-based copolymers show a single peak. A blue shift is observed as the Anthr content decreases, for example, the peak shifts from 545 nm (for AnthrCop 70/30 and AnthrHom) to 516 nm (for AnthrCop 10/90). This shift is due to reduced intermolecular interactions, resulting in emissions that are similar to those of the solution state.

For the white light-emitting terpolymers (WTPs), their solution photoluminescence profiles (Figure 1c) are dominated by the bimodal, peak owing to the Cz moiety at λ = 424–441 nm. For WTP1, a second shoulder ascribed to the Bz moiety at λ = 566 nm can be detected. On the other hand, for the terpolymers WTP2 and WTP3, no peak was detectable around 560 nm, owing to the low loading of these terpolymers on Bz. For the three white light-emitting terpolymers (WTPs), striking differences were observed in their emission profiles in film form (Figure 1f) compared to their emissions from solution (Figure 1c). Their solid-state emission is strongly dependent on the percentage of the Bz in the polymers. The WTP2 and WTP3 terpolymers with lower Bz loadings display emissions from both the Cz moiety and the Bz chromophore, indicating distinct energy transfer to Bz. Such energy transfer phenomena are ideal for achieving white light emission. On the other hand, WTP1, having the highest Bz loading, displays a dominant Bz emission at λ = 559 nm, with only a small emission at the lower wavelength arising from the Cz moiety (λ = 441 nm). It is evident that in the case of WTP1, the percentage of Bz is large enough to facilitate almost complete energy transfer from Cz to Bz and is therefore deficient for white light emission. Both WTP2 and WTP3, on the other hand, exhibit similar emissions for the blue and orange–red components. The emission of the **Cz** moiety is centered at around λ = 456 nm, and for the Bz moiety, experiencing a partial energy transfer from Cz, its emission is found at λ = 533 nm for WTP2 and at λ = 520 nm for WTP3. As seen, a difference of 0.5 mol% in the loading of Bz results in a different emission for the two copolymers, where the larger Bz loading of WTP2 results in a more efficient charge transfer from Cz, and as a result, the emission of Bz is dominant in the film of WTP2, whereas WTP3 displays a stronger blue emission. Moreover, a blueshift is observed when the content of Bz decreases in the copolymers WTP1 to WTP3. This phenomenon is due to the copolymeric film morphology. The lower mol% of Bz in WTP3 results in fewer intermolecular interactions.

To determine the HOMO and LUMO levels of the luminescent polymers, cyclic voltammograms were recorded for thin films deposited onto an FTO substrate. The oxidation potentials of all copolymers are presented in Figure 2. The HOMO level, which was irreversible in all cases, was determined from the onset of the oxidative wave using Equation (1). However, no clear determination of the LUMO level was possible via CV, and therefore, this was only by approximation determined from the absorption of the corresponding thin film using Equations (2) and (3). As is evident from Figure 2, the HOMO levels of the blue-emitting polymers are between −6.21 and −5.73 eV. The CzHom and CzCop 70/30 have HOMO levels at −6.21 eV and −6.07 eV, respectively. When the Cz moiety was incorporated in 10, 30, and 50 mol% in the polymeric chain, not much difference was seen in the HOMO levels. The LUMO levels for the polymers follow a similar trend, with CzCop 10/90, 30/70, and 50/50 having similar levels, while higher loadings of **Cz**, such as 70 and 100 mol%, have deeper and similar LUMO levels. In the case of the yellow-emitting polymers, they all exhibit similar energy levels, regardless of the chromophore loading on the polymeric chain. The HOMO levels for all Anthr-based polymers are between −5.86 eV (for AnthrCop 30/70) and, at the most, −5.7 eV (for AnthrHom). The LUMO levels are between −3.14 eV for AnthrHom and −3.31 eV for AnthrCops. Finally, for the white-emitting copolymers, WTP1 and WTP2 have similar energy levels, while WTP3 displayed deeper HOMO and LUMO levels. The detailed electrochemical data for all polymers that have been studied can be found in Appendix A.

### 3.4. Device Fabrication and Electroluminescence Characterization

As a next step, some of the studied polymers were implemented in PLED devices as emissive layers via solution deposition techniques. PLED devices were fabricated with the following simple structure: glass or PET/ITO/PEDOT:PSS (50 nm)/emissive layer/Ca (6 nm)/Ag (125 nm), as depicted in Appendix A. The PEDOT:PSS and the emissive layers were deposited via spin coating, for the case of glass substrates, or via slot die coating technique using a mini roller coater (FOM Technologies), for the case of PET substrates. The Ca and Ag layers were deposited via Vacuum Thermal Evaporation (VTE). The electroluminescence (EL) properties of these fabricated PLEDs were investigated to evaluate the emitting polymers’ suitability as emissive layers in stable, inexpensive, and solution-processable PLED devices. Spin coating was carried out inside gloveboxes. The fabrication and the characterization of the PLED devices were focused mainly on the possibility of applying the produced copolymers for the development of functional photoactive layers, the determination of emission characteristic profiles, and the evaluation of the stability of color emission through the evolution of the x-y CIE coordinates versus applied voltage. A thorough investigation of the fabrication parameters and device architectures to achieve enhanced performance of the PLED devices will be the next step in our future work.

#### 3.4.1. Blue and Yellow PLEDs

Appendix A displays the normalized intensities of EL emissions measured from the fabricated PLED devices containing the spin-coated CzHom, CzCop 50/50, and CzCop 10/90 films and the slot die-coated CzCop 50/50, as the emitting layers. Representative PLED devices of the Cz-based polymers are shown in Appendix A. According to Appendix A, the EL emission peak of CzHom is centered at 457 nm, and in the case of CzCop 50/50 and CzCop 10/90, the dominant EL peak is located at 456 nm. On the other hand, we observe that the slot die coated CzCop 50/50 EL emission shifts to a longer wavelength compared to spin coating. This red shift in the case of slot die-coated CzCop 50/50 may be due to an increase in charge traps, such as impurities or other structural defects, which create localized states between the HOMO and LUMO levels of the emissive layer. Another possible explanation could be that aggregation phenomena are more extended in the case of the slot die coating deposition technique due to the slower formation of the thin film and using a higher-concentration polymer solution than in the spin coating technique [11]. It is obvious that the emission band, which is obtained from the slot die coating device, is broader and without features. 

Moreover, the CIE chromaticity coordinates were also investigated, as derived from the EL measurements, and are depicted in Appendix A. The devices of CzHom, CzCop 50-50, and CzCop 10-90 exhibit CIE coordinates very close to the blue region; the respective values are (0.17, 0.17), (0.16, 0.16), and (0.17, 0.17). In the case of the slot die-coated CzCop 50-50, the CIE coordinates shifted to the sky-blue emission. Specifically, the CzCop 50-50_Slot-die obtains values at (0.24, 0.32).

To date, creating blue light-emitting materials with high color stability and selectivity poses a challenge for applications in PLED devices. This is due to their wide energy band gap, which leads to unbalanced injection and transportation of charges [46,47]. For this reason, to gain insight into the EL stability during operation, EL spectra were recorded at different applied voltages, as shown in Figure 3a–d. In our work, all EL spectra of the blue-emitting materials exhibit superior emission stability. 

It is also important to investigate the dependence of the CIE coordinates on voltage, as shown in Figure 3e,f. The CIE coordinates of the devices based on spin coating fabrication (Figure 3e) approach the ideal blue region and are independent of the applied driving voltages. On the other hand, the CIE coordinates of the PLED based on CzCop 50/50 via slot die coating emit in the sky-blue region. Figure 3f confirms the excellent color stability of the fabricated PLEDs during device operation, as no variations are observed in the CIE values. This fact indicates that the blue-emitting materials are promising candidates not only as an emissive layer in PLED devices but also as host materials for internal color conversion in blends with other conjugated polymers or copolymers. 

Table 3 summarizes the results of the operational and electrical characteristics of fabricated blue PLEDs. The characteristics curves of current density and luminance versus voltage are presented in Appendix A. The observed high turn-on voltage of the device based on CzCop 10/90 may be ascribed to the thin film morphology or thickness. Non-uniformity of the thin film introduces additional trap states, thereby reducing charge mobility and resulting in an elevated turn-on voltage. The operational characteristics of the fabricated devices present encouraging results, such as the luminance value reaching 130 cd/m^2^ from the CzCop 50/50 via slot die deposition, which could be applied for large-scale PLED production.

The EL spectra of the anthracene-based PLEDs obtained at 12 V are depicted in Appendix A, corresponding to the yellow region, as the EL maximum is centered at approximately 560 nm. Representative PLEDs of the anthracene-based polymers are shown in Appendix A. This fact is verified by the CIE coordinate diagram (Appendix A, showing that the CIE coordinates of all studied devices are in the yellow region. Specifically, the CIE coordinates of AnthrHom and AnthrCop 70/30 are equal to (0.43, 0.50) and (0.44, 0.50), respectively. In the case of AnthrCop 50/50 via spin coating, the coordinates calculated are (0.43, 0.51), while for the AnthrCop 50/50_Slot die, they are (0.47, 0.51).

A noticeable feature is the stability of the EL spectra when the voltage is increased for all the studied yellow PLEDs, as illustrated in Figure 4a–d. More specifically, increasing the applied voltage from 7 V to 12 V neither altered the shape nor the position of EL spectra. 

The evolution of the CIE coordinates derived from EL measurements is shown in Figure 4e,f. The results reveal that the emission of the studied devices is consistently in the yellow region. Figure 4f shows the evolution of EL CIE coordinates under different applied voltages. The color stability of all PLEDs studied is verified. The characteristics curves of current density and luminance versus voltage are depicted in Appendix A. The operational and electrical characteristics are summarized in Table 3. It is also worth mentioning that AnthrCop 70/30 demonstrates a strong potential for application to large-scale PLED devices for commercialization, as the fabricated PLED combined promising operational characteristics, for example, luminance value as high as 415 cd/m^2^, and it also exhibited pure and stable yellow emission.

#### 3.4.2. White PLEDs

Generally, for the WOLED technology, both the color quality and the stability have to be considered for the practical verification of the performance of OLEDs. Therefore, Figure 5 shows the evolution of the EL emission under different applied voltages for the white light-emitting terpolymers (a) WTP1, (b) WTP2, and (c) WTP3. This is a crucial factor since the increment in voltage usually increases the emitted light intensity, but on the other hand, this may end up causing unbalanced mixing of emitters [48,49,50]. In the WTPs examined here, it was found that increasing the applied bias led to an increase in the EL intensity. However, the positions of the maxima of the EL spectra remained constant and independent of the bias voltage. Additionally, the EL emission shows similar spectral shape to the PL emission profiles, as illustrated in Figure 5 versus Figure 1f. Specifically, in the case of WTP1, the EL emission is dominated by a peak at 554 nm, which originates from the Bz unit. Also, a weak emission located at 442 nm attributed to the Cz unit is observed, due to the incomplete energy transfer from Cz to Bz. On the other hand, one can see that both WTP2 (Figure 5b) and WTP3 (Figure 5c) devices exhibited broad emission covering the entire visible region, from 400 nm to 700 nm, which is produced by the combined emissions of blue and red chromophores. The EL emission of WTP2 (Figure 5b) presents approximately two balanced intensity peaks, located at 448 nm and 551 nm, respectively. This may be owing to the double contribution of incomplete energy transfer between the two chromophores and the direct charge-trapping of Cz. In the case of WTP3 (Figure 5c), the dominant peak is at 447 nm, and a weaker peak is located at 545 nm. This reinforced emission of blue chromophores may be attributed to the partial energy transfer between the two chromophores and more efficient charge trapping of the Cz unit. In the two latter cases, of WTP2 and WTP3, the combination of the sufficient emission of the blue unit as well as of the red unit leads to enhanced proximity to white light emission due to energy transfer. So, in the EL process, both energy transfer and carrier trapping phenomena take place. It is derived that the content of red chromophores must remain below 1% within the copolymer to achieve sufficient white light emission [2]. In addition, the characteristic curves of current density and luminance versus voltage are depicted in Figure 5d.

As mentioned above, to gain the desirable white emission, a broad emission covering the visible range is necessary. This fact can be verified by the CIE coordinates, as for the pure white light emission, the coordinates are (0.33, 0.33) [2,51,52]. Figure 5e displays the CIE chromaticity coordinates calculated from the EL measurements. The CIE coordinates of WTP1 are in the yellowish region, and more specifically, they are located at (0.40, 0.48), resulting from the Bz contribution that almost dominates the EL emission spectrum. When the concentration of Cz increased in the terpolymers, it is derived that the CIE coordinates shift to the white region. The CIE coordinates of WTP2 are (0.33, 0.38), and for WTP3, they are (0.29, 0.31). The evolution of (x, y) coordinates with the applied voltage confirmed the superior color stability of the devices, as shown in Figure 5f. So, it is verified that the white light-emitting terpolymers (WTPs) exhibit superior color stability without significant variations during the device operation, which is a key factor for lighting applications.

These systematic studies were conducted to evaluate the potential of these terpolymers as emissive layers in WOLED devices, with a primary focus on the color emission characteristics. Concerning their indicative operational characteristics, these are presented in Table 4. It is important to mention that promising CRI values are obtained. In the case of WTP3, the CRI value reached 83 and the CCT value was equal to 8000 K. So, encouraging results are demonstrated for achieving white light emission with color stability, with very promising CRI and luminance values. Clearly, further device optimization is required to improve the performance of the produced PLED devices.

Ultimately, when human-centric lighting technologies are considered, such fine hue tuning of the white light emission is advantageous over complex multicomponent lighting device architectures. 

## 4. Conclusions

In conclusion, in this study, fluorescent poly (arylene ether sulfone)s were explored as the EML of PLEDs. Blue-, yellow-, and orange–red-emitting chromophores were utilized to create polymers that emit blue, yellow, or red light, or white light by carefully adjusting the ratio of chromophores. Single-batch polymer quantities up to 25 g were successfully produced, with Mn even above 70 KDa and high thermal stability. The high-temperature polymerization protocol employed does not involve the use of expensive rare metal catalysts, as in typical fully conjugated photoactive polymers, significantly lowering the materials’ production cost and eliminating metal catalyst traces, which, for complete removal, requires tedious purification procedures. The fluorescent polysulfones presented herein showed excellent solubility at room temperature in common non-chlorinated, non-volatile solvents, produced stable inks, even in concentrations as high as 10%wt, and can be printed or casted from their solutions, verifying the versatility of poly (arylene ether)s as a framework to develop materials for the EML of large-area, printed, and flexible PLED devices, with minimal differences between batches. 

In all cases, the synthesized poly(arylene ether)s were found to be suitable for use in the EML of PLEDs. Results from PLEDs show superior color stability, and for the case of WTPs, exceptional broadband EL emission covering the visible range, accompanied by promising CRI values and stable color in the white region under different applied voltages.

## Data Availability

Experimental data are available from the authors upon reasonable request.

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
