# Peer review of "Fluorescent Aromatic Polyether Sulfones: Processable, Scalable, Efficient, and Stable Polymer Emitters and Their Single-Layer Polymer Light-Emitting Diodes"

_nanomaterials, 2024, doi:10.3390/nano14151246_

Round 1

Reviewer 1 Report

Comments and Suggestions for Authors

The manuscript investigates the preparation and application of novel fluorescent poly(arylene ether sulfone)s for use as emissive layers in polymer light-emitting diodes (PLEDs). The study highlights the integration of blue, yellow, and orange-red emitting chromophores into polymers that exhibit high molecular weights, excellent solubility, and thermal stability. Additionally, the paper reports the successful fabrication of PLED devices using these polymers, which demonstrate superior color stability and promising luminance values. This manuscript is a solid contribution to the field of luminescent materials, particularly for applications in PLEDs. With minor revisions, the paper will be suitable for publication in Nanomaterials.

1. Line 26-27: "the fluorescent copolyethers deposition" should be "the deposition of fluorescent copolyethers".

2. Line 44-45: "These advantages mainly being lightweight, comparatively simple device structure, cost effective and ease of fabrication, printable, requiring a single emitting layer." needs restructuring for clarity. Consider: "These advantages include being lightweight, having a comparatively simple device structure, being cost-effective and easy to fabricate, being printable, and requiring only a single emitting layer."

3. Line 101-104: "A diphenylpyridine comonomer was further used alongside the fluorescent monomers in copolymeric or terpolymeric structures that improved further the solubility of the materials while enhancing their fluorescence properties and color purity by diluting and separating adjacent chromophores." could be clearer as "A diphenylpyridine comonomer was also incorporated alongside the fluorescent monomers in copolymeric or terpolymeric structures, which further improved the solubility of the materials and enhanced their fluorescence properties and color purity by increasing the distance between adjacent chromophores."

Author Response

Response to the Reviewer 1 comments regarding the Manuscript nanomaterials-3055469

The manuscript investigates the preparation and application of novel fluorescent poly(arylene ether sulfone)s for use as emissive layers in polymer light-emitting diodes (PLEDs). The study highlights the integration of blue, yellow, and orange-red emitting chromophores into polymers that exhibit high molecular weights, excellent solubility, and thermal stability. Additionally, the paper reports the successful fabrication of PLED devices using these polymers, which demonstrate superior color stability and promising luminance values. This manuscript is a solid contribution to the field of luminescent materials, particularly for applications in PLEDs.

With minor revisions, the paper will be suitable for publication in Nanomaterials.

The authors would like to thank the reviewer for the careful assessment of our work and the positive feedback.

Below are our responses point by point to the reviewer’s specific comments.

  1. Line 26-27: "the fluorescent copolyethers deposition" should be "the deposition of fluorescent copolyethers".

Response: The correction has been incorporated in the text. Thank you.

  1. Line 44-45: "These advantages mainly being lightweight, comparatively simple device structure, cost effective and ease of fabrication, printable, requiring a single emitting layer." needs restructuring for clarity. Consider: "These advantages include being lightweight, having a comparatively simple device structure, being cost-effective and easy to fabricate, being printable, and requiring only a single emitting layer."

Response: The revised sentence has been used. Thank you.

  1. Line 101-104: "A diphenylpyridine comonomer was further used alongside the fluorescent monomers in copolymeric or terpolymeric structures that improved further the solubility of the materials while enhancing their fluorescence properties and color purity by diluting and separating adjacent chromophores." could be clearer as "A diphenylpyridine comonomer was also incorporated alongside the fluorescent monomers in copolymeric or terpolymeric structures, which further improved the solubility of the materials and enhanced their fluorescence properties and color purity by increasing the distance between adjacent chromophores."

Response: The rephrased sentence has been adopted. We appreciate your help and support.  

Reviewer 2 Report

Comments and Suggestions for Authors

The authors have developed fully aromatic polyether sulfones with blue, yellow, and orange-red π-conjugated semiconducting units. Additionally, they have successfully tuned white-emitting copolymers by making subtle changes to the comonomers' type and ratio. The manuscript, however, lacks specific details regarding the fabrication and characterization of the PLEDs.

1. The authors should describe these processes in detail, especially if they were carried out in gloveboxes to prevent exposure to oxygen and water. If this precaution was not taken, it could potentially explain the poor device performance observed.

2. In order to justify the colorimetry data, it is crucial to provide clear EL spectra. Since the PLED with a Ca cathode quickly degraded in the presence of water and oxygen, I highly recommend that the authors fabricate and characterize the device within a proper glovebox environment to ensure accurate and reliable results.

3. To evaluate the performance of the PLEDs comprehensively, it is essential to include basic device characteristics such as luminance-current-voltage and efficiency-current curves. These parameters will provide valuable insights into their overall performance and efficiency.

Author Response

Response to the Reviewer 2 comments regarding the Manuscript nanomaterials-3055469

The authors have developed fully aromatic polyether sulfones with blue, yellow, and orange-red π-conjugated semiconducting units. Additionally, they have successfully tuned white-emitting copolymers by making subtle changes to the comonomers' type and ratio. The manuscript, however, lacks specific details regarding the fabrication and characterization of the PLEDs.

The authors would like to thank the reviewer for the careful assessment of our work and the constructive feedback.

Below are our responses point by point to the reviewer’s specific comments.

  1. The authors should describe these processes in detail, especially if they were carried out in gloveboxes to prevent exposure to oxygen and water. If this precaution was not taken, it could potentially explain the poor device performance observed.

Response: The authors thank the reviewer for this insightful comment. We have rewritten the respective paragraph (section 3.4) giving all the appropriate details concerning the fabrication processes of the devices.

  1. In order to justify the colorimetry data, it is crucial to provide clear EL spectra. Since the PLED with a Ca cathode quickly degraded in the presence of water and oxygen, I highly recommend that the authors fabricate and characterize the device within a proper glovebox environment to ensure accurate and reliable results.

Response: We have rewritten the respective paragraph (section 3.4) giving all the appropriate details concerning the environment (glove box) into which the EL measurements were performed.

In this work, all deposition processes, including both spin coating and slot-die coating, were conducted in the glove box environment. It is also important to mention that we focus on the design, synthesis, and comprehensive investigation of the optical, photophysical, and electro-optical properties of novel emitting materials, with the aim of utilizing them as emitting layers in large-scale, printed OLED devices. Indeed, the measured luminance of the OLED devices is relatively low but this can be ascribed mainly to the low thickness of the emissive polymers. Additionally, the fabrication process for OLED devices has not yet been fully optimized. Future improvements will include fine-tuning the device architecture to ensure that each layer's energy levels are properly aligned, which is expected to significantly enhance device efficiency.

  1. To evaluate the performance of the PLEDs comprehensively, it is essential to include basic device characteristics such as luminance-current-voltage and efficiency-current curves. These parameters will provide valuable insights into their overall performance and efficiency.

Response: We agree with the referee. For this reason, we added the characteristics curves of Current Density and Luminance versus Voltage in the Supporting Information file for the blue and yellow PLEDs Figures S16c and S17c, respectively, and in the new Figure 5d of the revised manuscript for the WTPs.

Reviewer 3 Report

Comments and Suggestions for Authors

The manuscript under review is clear, comprehensive, and highly relevant to the field. The authors successfully develop Polymer Emitters and their Single Layer Polymer Light Emitting Diodes, providing a compelling rationale for the study and emphasizing its importance.

Despite the presence of similar papers published recently, this manuscript remains relevant and of significant interest to the scientific community. The author effectively presents new perspectives and insights that distinguish this work from previous studies.

The cited references are predominantly recent publications within the last five years and are highly pertinent to the topic. The author has meticulously included key studies, ensuring a thorough and up-to-date literature review. There are no noticeable omissions of relevant citations, and the manuscript avoids an excessive number of self-citations, which maintains the objectivity and credibility of the paper.

The statements and conclusions drawn are coherent and well-supported by the listed citations. The author skillfully synthesizes information from multiple sources, leading to well-founded conclusions that enhance our understanding of the subject.

The figures, tables, images, and schemes are appropriate and effectively illustrate the data. They are clear, easy to interpret, and complement the text well, thereby enhancing the overall readability and impact of the manuscript.

In conclusion, this manuscript makes a valuable contribution to the field. It is well-researched, clearly written, and provides new results into PLEDs. I highly recommend it for publication and believe it will be a valuable resource for researchers.

In Figure S14, it is unclear why the absorbances are simply normalized to the maximum value. It would be clearer to normalize to the absorbance peaks in the blue and green regions in a), b), and c). This adjustment would make the graphs more intuitive and help readers better compare the data under different conditions.

Additionally, graph d) is difficult to interpret with this normalization. The same observation applies to graphs e) and f). Normalizing based on specific peaks would make the data more interpretable and provide a clearer picture of the spectroscopic variations.

This modification would significantly improve the clarity and understanding of the presented results

I suggest changing the y-axis title from "N. Absorbance (a.u.)" to "Absorbance (a.u.)" and specifying in the caption how the normalizations were performed.

In figure 1 I suggest changing the y-axis title from "N. PL Intensity (a.u.)" to " PL Intensity (a.u.)" and specifying in the caption how the normalizations were performed

I suggest moving the photographs from Figure 1 to the supplementary materials, making the figures simpler and easier to read

Author Response

Response to the Reviewer 3 comments regarding the Manuscript nanomaterials-3055469

The manuscript under review is clear, comprehensive, and highly relevant to the field. The authors successfully develop Polymer Emitters and their Single Layer Polymer Light Emitting Diodes, providing a compelling rationale for the study and emphasizing its importance. Despite the presence of similar papers published recently, this manuscript remains relevant and of significant interest to the scientific community. The author effectively presents new perspectives and insights that distinguish this work from previous studies. The cited references are predominantly recent publications within the last five years and are highly pertinent to the topic. The author has meticulously included key studies, ensuring a thorough and up-to-date literature review. There are no noticeable omissions of relevant citations, and the manuscript avoids an excessive number of self-citations, which maintains the objectivity and credibility of the paper. The statements and conclusions drawn are coherent and well-supported by the listed citations. The author skillfully synthesizes information from multiple sources, leading to well-founded conclusions that enhance our understanding of the subject. The figures, tables, images, and schemes are appropriate and effectively illustrate the data. They are clear, easy to interpret, and complement the text well, thereby enhancing the overall readability and impact of the manuscript. In conclusion, this manuscript makes a valuable contribution to the field. It is well-researched, clearly written, and provides new results into PLEDs.

I highly recommend it for publication and believe it will be a valuable resource for researchers.

The authors would like to thank the reviewer for the careful assessment of our work and the positive feedback.

Below are our responses point by point to the reviewer’s specific comments.

  1. In Figure S14, it is unclear why the absorbances are simply normalized to the maximum value. It would be clearer to normalize to the absorbance peaks in the blue and green regions in a), b), and c). This adjustment would make the graphs more intuitive and help readers better compare the data under different conditions. Additionally, graph d) is difficult to interpret with this normalization. The same observation applies to graphs e) and f). Normalizing based on specific peaks would make the data more interpretable and provide a clearer picture of the spectroscopic variations.This modification would significantly improve the clarity and understanding of the presented results. I suggest changing the y-axis title from "N. Absorbance (a.u.)" to "Absorbance (a.u.)" and specifying in the caption how the normalizations were performed.

Response: In Figure S14, the y-axis has been changed from N.Absorbance (a.u.) to Absorbance (a.u.) as requested by the reviewer. In the figure caption, it is now specified that normalizations have been performed at the 275nm peak in all cases. The reasoning behind this choice is to use a common peak of unaffected intensity by the chromophore content. This way the percentage of the blue, green, and reddish chromophores is depicted clearly with their respective peaks’ intensities. As seen in Figure S14a for example, the percentage of the carbazole chromophore increasing from 10% to 100% in the CzCop10/90 to the CzHom, leads to an obvious intensity increase of the peak centered at 375 nm owing to the carbazole absorption. If normalization took place at the blue peak then no differentiations would be noticeable which would be misleading.

  1. In figure 1 I suggest changing the y-axis title from "N. PL Intensity (a.u.)" to " PL Intensity (a.u.)" and specifying in the caption how the normalizations were performed.

I suggest moving the photographs from Figure 1 to the supplementary materials, making the figures simpler and easier to read

Response: In Figure 1, the y-axis has been changed from N. PL Intenstiy (a.u.) to PL Intenstiy (a.u.) as requested by the reviewer. In the figure caption it is specified that the spectra were normalized at the emission maxima. Also, in order to follow the reviewer’s suggestion regarding the photographs, these are now placed next to the respective graphs, so as to be directly visible to the readers but not to interfere with the photoluminescence curves.

Reviewer 4 Report

Comments and Suggestions for Authors

In this manuscript, the authors developed a series of fully aromatic polyether sulfones emitting blue, yellow and white light by incorporating carbazole, anthracene and benzothiadiazole as the fluorophores, respectively. By utilizing the synthetic methodology, the copolymers can be prepared in large scale up to 25 g with high molecular weight and thermal stability. The data is reliable and interesting. However, in-depth discussions about the photophysical properties and EL performance of the copolymers are needed. In my opinion, the manuscript can be considered to be accepted after addressing the following issues.

1. As an electron acceptor, what kind of role did diphenylsulfone play in the copolymers, especially in CzHom and CzCop series with carbazole as the typical electron donor? No discussion about the potential charge transfer process can be found.

2. The fluorescence quantum yields of the copolymers by varying the feed ratio of Hopy should be provided.

3. In Figure 1a and 1b, why the relative intensities between the two peaks are proportionate to the percentage of carbazole in the polymeric chain?

4. In Figure 1e, why the emitting wavelength showed a blue shift with decreasing the Anthr content?

5. In Figure 1f, why the emission from Bz moiety exhibited blue shift with decreasing the content of Bz?

6. In Figure 2d and 2f, why the HOMO energy level was decreased with increasing the loading ratio of Cz, which is an electron-donating moiety?

7. The EL performance of the copolymers is unsatisfying. The luminance values are relatively low, and some key parameters such as current efficiency, power efficiency and external quantum efficiency, should be provided.

8. The turn-on voltage of the device based on CzCop10/90 is extremely high. The probable reason should be discussed.

Comments on the Quality of English Language

The quality of English language is fine.

Author Response

Response to the Reviewer 4 comments regarding the Manuscript nanomaterials-3055469

In this manuscript, the authors developed a series of fully aromatic polyether sulfones emitting blue, yellow and white light by incorporating carbazole, anthracene and benzothiadiazole as the fluorophores, respectively. By utilizing the synthetic methodology, the copolymers can be prepared in large scale up to 25 g with high molecular weight and thermal stability. The data is reliable and interesting. However, in-depth discussions about the photophysical properties and EL performance of the copolymers are needed. In my opinion, the manuscript can be considered to be accepted after addressing the following issues.

The authors would like to thank the reviewer for the careful assessment of our work and the valuable indications.

Below are our responses point by point to the reviewer’s specific comments.

  1. As an electron acceptor, what kind of role did diphenylsulfone play in the copolymers, especially in CzHom and CzCop series with carbazole as the typical electron donor? No discussion about the potential charge transfer process can be found.

Response: As mentioned in the main text, lines 83, 247-249, our main reasoning behind choosing diphenylsulfone as a comonomer for all copolymers mainly stemmed from the fact that it affords high molecular weight soluble polyethers, with high thermal, chemical and oxidative stability. We reasoned that it could also impart good charge transport properties to the films due to its electron-accepting nature. Diphenylsulfone being at the centre of TADF-based small molecule emitters could also enhance the emission of the copolymers or even give rise to exciplexes during operation. We are in the process of further investing in these aspects for our copolymers.

  1. The fluorescence quantum yields of the copolymers by varying the feed ratio of Hopy should be provided.

Response: Indeed, as the reviewer suggested, the PL efficiencies of the materials could give additional information on their optoelectronic properties. Unfortunately, though, the required instrumentation, PL with an integrating sphere, is not available nor at reach by the authors. Therefore, we were forced to proceed with the electroluminescence properties evaluation of our copolymers without their QE assessment.

  1. In Figure 1a and 1b, why the relative intensities between the two peaks are proportionate to the percentage of carbazole in the polymeric chain?

Response: The difference in the relative intensities of the emission peaks could be attributed to the different interactions of the chromophore (Cz and Anthr) moieties in the solution state. Since in the film state, all copolymers show an amorphous emission profile, we could arrive at an early conclusion that in the solution state, the moieties have the necessary freedom to interact, given the dynamic nature of the system, that is absent in the film state. However, more in-depth, time-dependent spectroscopic investigations are needed to corroborate this rough explanation.

This point has been incorporated in the main text as well, in Lines 369-373 and in Lines 378-379.

  1. In Figure 1e, why the emitting wavelength showed a blue shift with decreasing the Anthr content?

Response: In the solid state, with decreasing chromophore content, there are fewer intermolecular interactions, leading to an emission that resembles that of the solution state. For this reason, in the solid state the emissions of the copolymers with less Anthr contents have λmax closer to 500 nm, which is the λmax of the solution state for all copolymers.

This point has been incorporated in the main text as well, in Lines 384-385.

  1. In Figure 1f, why the emission from Bz moiety exhibited blue shift with decreasing the content of Bz?

Response: As mentioned above, increasing the chromophores contents leads to increased intermolecular interactions in the solid state and thus the emission profiles are redshifted compared to those of solutions. Therefore, the emission curves of the WPT1>WTP2>WTP3 with 5%, 0.5%, 0.25% ratios of the Bz chromophore, respectively, have redshifted emissions following the same order. The explanation has been incorporated in the main text in lines 424-427.

  1. In Figure 2d and 2f, why the HOMO energy level was decreased with increasing the loading ratio of Cz, which is an electron-donating moiety?

Response: In all cases, the cyclic voltammograms of Figure 2 were recorded for films deposited onto FTO substrates via drop casting. This deposition technique leads to inevitable deviations of the films’ thickness and uniformity, and segregation of the chromophores within the deposited films due to slow drying. Thus, in several cases, discrepancies from the expected behaviour of the polymers' energy levels occurred, such as in the cases of the CzHom and CzCop 70/30. Nonetheless, these discrepancies happen in few cases and the deviations lie within experimental errors.

  1. The EL performance of the copolymers is unsatisfying. The luminance values are relatively low, and some key parameters such as current efficiency, power efficiency and external quantum efficiency, should be provided.

Response: We agree with the referee. For this reason, we added the characteristics curves of Current Density and Luminance versus Voltage in the Supporting Information file for the blue and yellow PLEDs, Figures S16c and S17c respectively, and in the new Figure 5d of the revised manuscript for the WOLEDs.

Indeed, the observed luminance values are relatively low. This study is primarily focused on the synthesis and emission characteristics of the developed emitting materials. The characteristics of devices based on high operating efficiency such as current efficiency, power efficiency, and EQE are our future goal. In particular, this research focuses on the importance of solution-processed OLED devices based on the synthesis of new materials as emissive layers. These devices were initially fabricated using the spin coating technique and later transferred to slot-die coating, which is compatible with Roll-to-Roll production. The findings demonstrate promising outcomes across all studied cases, particularly regarding the printability of the synthesized materials and the operational stability of devices. Continued research and improvement in device efficiency could significantly advance the field. Thus, this study is a significant step forward, highlighting the possibility of device manufacturing based on the production of large-area lighting devices due to its cost-effectiveness and simple manufacturing processes. Thus, for the readers to have a clearer statement on the aim of this work on the device level, we have added the below comment in new Lines 468-474, in the revised manuscript:

“The fabrication and the characterization of the PLED devices were focused mainly on the possibility of applying the produced copolymers for the development of functional photoactive layers, the determination of emission characteristic profiles, and the evaluation of the stability of colour emission through the evolution of the x-y CIE Coordinates versus applied voltage. A thorough investigation of the fabrication parameters and devices’ architectures to achieve enhanced performance of the PLEDs devices will be the next step in our future work.”

  1. The turn-on voltage of the device based on CzCop10/90 is extremely high. The probable reason should be discussed.

To clarify this issue, also for the readers, we have added the following sentence in the revised manuscript, in lines 515 to 519:

“The observed high turn-on voltage of the device based on CzCop 10/90, may be ascribed to the thin film morphology or thickness. Non-uniformity of the thin film introduces additional trap states, thereby reducing charge mobility and resulting in an elevated turn-on voltage.”

Round 2

Reviewer 2 Report

Comments and Suggestions for Authors

Issues have been properly addressed.

Author Response

The authors would like to thank the Reviewer for his/her concern.

Reviewer 4 Report

Comments and Suggestions for Authors

In this version, the authors addressed most of the issues listed in the comments.  However, some answers are still unclear, and the manuscript can not be published at this stage. The questions are listed as follows:

1. For question 6, the authors explained that “Thus, in several cases, discrepancies from the expected behaviour of the polymers' energy levels occurred, such as in the cases of the CzHom and CzCop 70/30. Nonetheless, these discrepancies happen in few cases and the deviations lie within experimental errors.” If the results only reflect the discrepancies rather than the true fact, the discussion “with larger Cz percentages having a deeper HOMO level (line 436)” must be revised.

2. For question 8, the authors answered “The observed high turn-on voltage of the device based on CzCop 10/90, may be ascribed to the thin film morphology or thickness. Non-uniformity of the thin film introduces additional trap states, thereby reducing charge mobility and resulting in an elevated turn-on voltage.” The extraordinary high turn-on voltage was derived from the quality of the film, which was highly depended on the fabrication technic. As a result, experiments of repeatability and the corresponding results are needed.

Comments on the Quality of English Language

The quality of English is fine.

Author Response

In this version, the authors addressed most of the issues listed in the comments.  However, some answers are still unclear, and the manuscript can not be published at this stage. The questions are listed as follows:

The authors would like to thank the reviewer for the careful assessment of our work and the valuable indications.

Below are our responses point by point to the reviewer’s specific comments.

  1. For question 6, the authors explained that “Thus, in several cases, discrepancies from the expected behaviour of the polymers' energy levels occurred, such as in the cases of the CzHom and CzCop 70/30. Nonetheless, these discrepancies happen in few cases and the deviations lie within experimental errors.” If the results only reflect the discrepancies rather than the true fact, the discussion “with larger Cz percentages having a deeper HOMO level (line 436)” must be revised.

Response: This part has been revised as directed by the reviewer. In Lines 435-438 a new sentence has been incorporated “The CzHom and CzCop 70/30 have HOMO levels at -6.21 eV and -6.07 eV, respectively, while when the Cz moiety was incorporated in 10, 30 and 50 mol% in the polymeric chain, not much difference was seen in the HOMO levels.”

  1. For question 8, the authors answered “The observed high turn-on voltage of the device based on CzCop 10/90, may be ascribed to the thin film morphology or thickness. Non-uniformity of the thin film introduces additional trap states, thereby reducing charge mobility and resulting in an elevated turn-on voltage.” The extraordinary high turn-on voltage was derived from the quality of the film, which was highly depended on the fabrication technic. As a result, experiments of repeatability and the corresponding results are needed.

Response: In the 1st revised version of the manuscript we have added (page 11 Lines 468-474) the following statement:

“The fabrication and the characterization of the PLED devices were focused mainly on the possibility of applying the produced copolymers for the development of functional photoactive layers, the determination of emission characteristic profiles, and the evaluation of the stability of colour emission through the evolution of the x-y CIE Coordinates versus applied voltage. A thorough investigation of the fabrication parameters and devices’ architectures to achieve enhanced performance of the PLEDs devices will be the next step in our future work.”

as an answer to comment #7 of the first Reviewer Report.

Thus, the optimization of the devices’ performances, which includes also the achievement of low turn-on voltages, is clearly beyond the aims of this paper. Furthermore, the nominal value of the voltage does not affect the emission characteristics, whereas the desired stability of the emission characteristics is achieved and certified by the derived results of the evolution of CIE coordinates under different applied voltages (Fig. 3e).

Ensuring these in this work, enables us to proceed further in the future to the detailed investigation on the enhancement of the devices' performance by improving the films' quality, as well as the investigation of the proper devices' architectures. So, we strongly believe that the additional experiments for the verification of the repeatability of the fabrication technique are not necessary to be presented in this work, as this is dedicated to the synthesis of  Processable, Scalable, Efficient, and Stable Polymer Emitters.

Round 3

Reviewer 4 Report

Comments and Suggestions for Authors

In this version, the authors addressed the issues mentioned, and the manuscript can be accepted in Nanomaterials.

Comments on the Quality of English Language

The quality of English is acceptable.

Author Response

(The authors gave the same response as above.)
